# Expansion of cytotoxic tissue-resident CD8+ T cells and CCR6+CD161+ CD4+ T cells in the nasal mucosa following mRNA COVID-19 vaccination

Aloysious Ssemaganda [1], Huong Mai Nguyen[1], Faisal Nuhu[1], Naima Jahan[1], Catherine M. Card [1,2], Sandra Kiazyk[1,2], Giulia Severini[1], Yoav Keynan[1], Ruey-Chyi Su[1,2], Hezhao Ji[1,2], Bernard Abrenica[2], Paul J. McLaren [1,2], T. Blake Ball[1,2], Jared Bullard[1,3,4], Paul Van Caeseele[1,3], Derek Stein[1,3] & Lyle R. McKinnon[1,2,5 ✉]

Vaccines against SARS-CoV-2 have shown high efficacy in clinical trials, yet a full immunologic characterization of these vaccines, particularly within the human upper respiratory tract, is less well known. Here, we enumerate and phenotype T cells in nasal mucosa and blood using flow cytometry before and after vaccination with the Pfizer-BioNTech COVID-19 vaccine ($n = 21$). Tissue-resident memory (Trm) CD8+ T cells expressing CD69+CD103+ increase in number ~12 days following the first and second doses, by 0.31 and 0.43 $\log_{10}$ cells per swab respectively ($p = 0.058$ and $p = 0.009$ in adjusted linear mixed models). CD69+CD103+CD8+ T cells in the blood decrease post-vaccination. Similar increases in nasal CD8+CD69+CD103− T cells are observed, particularly following the second dose. CD4+ cells co-expressing CCR6 and CD161 are also increased in abundance following both doses. Stimulation of nasal CD8+ T cells with SARS-CoV-2 spike peptides elevates expression of CD107a at 2- and 6-months ($p = 0.0096$) post second vaccine dose, with a subset of donors also expressing increased cytokines. These data suggest that nasal T cells may be induced and contribute to the protective immunity afforded by this vaccine.

[1] Department of Medical Microbiology and Infectious Diseases, University of Manitoba, Winnipeg, MB, Canada. [2] JC Wilt Infectious Diseases Research Centre, National Microbiology Laboratory, Public Health Agency of Canada, Winnipeg, MB, Canada. [3] Cadham Provincial Laboratory, Winnipeg, MB, Canada. [4] Department of Pediatrics & Child Health, University of Manitoba, Winnipeg, MB, Canada. [5] Centre for the AIDS Programme of Research in South Africa (CAPRISA), Durban, South Africa. ✉email: lyle.mckinnon@umanitoba.ca

 1

The COVID-19 pandemic has led to substantial mortality and caused major global disruption. At least six vaccines have demonstrated efficacy in preventing severe disease associated with SARS-CoV-2 infection[1]. While this is an impressive achievement guided by early immunogenicity data, a full understanding of how these novel vaccines interact with the host immune system continues to emerge. This is particularly the case in tissues such as the upper respiratory tract (URT), which are often more challenging to sample than peripheral blood[2]. Previous work has suggested influenza-specific CD8+ Tissue-resident memory (Trm) cells in the lung are responsive but wane with age[3]. In SARS-CoV-2 infection, lung Trm cells derived from sampling tissues obtained from surgery persisted for up to 10 months following recovery, suggesting that these cells contribute to protective immunity[4]. Here we focus on characterizing nasal mucosal T cells in healthy individuals pre- and post-vaccination with the BNT162b2 mRNA-nanoparticle vaccine developed by Pfizer-BioNTech[5].

Natural history studies of SARS-CoV-2 infection demonstrate significant cell-mediated and humoral immune responses[6–9], which in combination lead to a substantial decrease in re-infection risk[10]. These responses can be readily detected for >8 months post-infection[11], and appear to have a long half-life. In non-human primates, CD8+ T cell depletion led to increased susceptibility to re-infection[12]. Studies that assess the quality and quantity of immune responses induced by vaccination in humans are beginning to emerge[11–16], adding depth to what was generated in early phase clinical trials. While these studies suggest a high magnitude of humoral and cell-mediated immunity to SARS-CoV-2[13,14], less is known regarding tissue and mucosal immunity induced by these vaccines, particularly in the URT, the primary site of viral entry.

In this study, we demonstrate that there is an increase in the abundance of memory T cells with a tissue-resident phenotype within the nasal mucosa following vaccination with BNT162b2 mRNA vaccine. These cells may contribute to frontline immunity to SARS-CoV-2 at the time of virus exposure.

## Results and discussion

We developed an ex vivo flow cytometry-based assay to enumerate and profile immune cells isolated from nasopharyngeal (NP) swabs used for SARS-CoV-2 diagnostic testing amongst healthy volunteers. We pilot-tested several types of NP swabs to gauge optimal immune cell recovery. Strikingly, CD45+ immune cells were recovered from a certain type of swab, Flexible minitip flocked swab from BD Biosciences, while CD326+ epithelial cells were the main cell type recovered from two other swabs that we tested, including the Copan FLOQ swab (Fig. 1A, B). In the pilot study ($n = 8$), a median of 3082 (IQR: 2351–6168) CD45+ cells were recovered per BD swab, ~80% of which were CD3+ T cells. Using CD69 and CD103 as markers of tissue-resident memory (Trm) T cells[15,16], virtually all CD8+ (>90%) and variable proportions of CD4+ T cells were Trm (10–80%; Fig. 1C). As described, a sizeable proportion of CD69+ CD103− CD4+ T cells in tissue may also be Trm[17].

We enrolled healthy volunteers scheduled to receive the BNT162b2 mRNA vaccine ($n = 21$, Supplementary Table 1), and collected baseline, pre-vaccine samples followed by peak response samples ~12 days after the first (visit 2) and second (visit 3) doses. Participants were a median age of 40 (IQR: 31–50) and predominantly female (14/21). Most participants (19/21) received influenza vaccination in the previous few months, and two participants had prior COVID-19 infection. The median body mass index (BMI) was 24 (IQR: 22–29). Reporting of underlying medical conditions included allergies and/or asthma ($n = 6$), type

II diabetes ($n = 2$), hypertension ($n = 2$), and autoimmunity ($n = 1$).

We analyzed both frequency of parent populations and total cell abundance, the latter by maximizing acquisition of all cells from each swab; measuring abundance is critical as cell abundance can vary within mucosa across orders of magnitude[18,19]. At baseline, a similar number of immune cells were retrieved as observed in pilot data in Fig. 1B, from a similar participant population, suggesting consistency of this sampling method. We observed significant increases in nasal CD8+ Trm cells at both peak vaccine response time points (Fig. 2A). In a linear mixed model adjusted for age, sex, BMI, prior COVID-19, influenza vaccination, and self-reported vaccine side effects, the number of CD8+ Trm increased by 0.31 and 0.44 $\log_{10}$ cells per nasal swab following the first and second vaccine dose, respectively ($p = 0.058$ and $p = 0.009$, adjusted linear mixed models). No increases in CD4+ Trm cells defined by CD103 and/or CD69 expression were observed (Fig. 2C). The absolute number of nasal CD8+CD69+CD103− T cells also increased at visit 2 and particularly at visit 3, by 0.06 and 0.48 $\log_{10}$ cells/swab, respectively (adjusted $p = 0.7$ and $p = 0.004$). The proportion of nasal CD8 + T cells that were Trm also increased, although not significantly ($p = 0.09$, Supplementary Fig. 1A, B). It should be noted that most pre-vaccine nasal CD8+ T cells already express a Trm phenotype. Interestingly, the low proportion of CD69+CD103+ T cells in the blood significantly declined at the peak vaccine response time points; one explanation for these data is that these cells may be migrating out of the blood into tissues (Fig. 2B–D). We also observed that CD4+ T cells co-expressing the Th17 markers CD161 and CCR6 increased by 0.40 and 0.45 $\log_{10}$ cells/nasal swab at the peak vaccine response time points compared to baseline (adjusted $p = 0.017$ and $p = 0.008$, respectively, Fig. 2E). No increases in nasal T follicular helper (Tfh) cells were observed following either vaccine dose (Fig. 2F). Similar increases in nasal CD161+CCR6+CD4+ T cells and CD8 Trm cells were observed in analyses stratified by age and sex (Supplementary Table 2), and excluding participants with prior COVID-19 infection did not meaningfully change any outcomes.

Studies of other vaccines suggest that peak vaccine-elicited T cells can be captured by activation markers such as Ki67, HLA-DR, and CD38, which peak at 10-14 days following yellow fever and smallpox vaccination[20]. In our study, we did not observe any changes in the proportion of CD4+ or CD8+ T cells co-expressing HLA-DR and CD38, or any upregulation of Ki-67, in either nasal or blood samples (Supplementary Fig. 2).

We further analysed our flow data using the t-distributed stochastic neighbor embedding (tSNE), a dimensionality reduction method. Several substantial shifts in CD45+ nasal immune cell clusters were evident from baseline to visit 2, and from visit 2 to 3 (Supplementary Fig. 3A). These included a step-wise shift in CD8+ T cell populations across the three time points, but also shifts in CD3− and CD3+CD4−CD8− cells. In support of this, the abundance of CD45+, CD3+, and CD8+ T cells all increased at visit 3, compared to baseline (Supplementary Fig. 3B–G). While it was difficult to distinguish which phenotypic markers in the flow panel explained the observed divergent clustering patterns, these data suggest that multiple subsets of nasal immune cells may increase in number following COVID-19 vaccination.

To assess the antigen-specificity of nasal T cells, we initially enrolled vaccinated individuals at 2 months following their second Pfizer-BNT dose, and either stimulated the entirety of the NP swab cells for 4 h with overlapping SARS-CoV-2 spike protein peptide pools ($n = 15$ participants), or left cells in media unstimulated ($n = 7$). Stimulated samples had increased expression of the cytotoxic marker, CD107a ($p = 0.01$, Unpaired

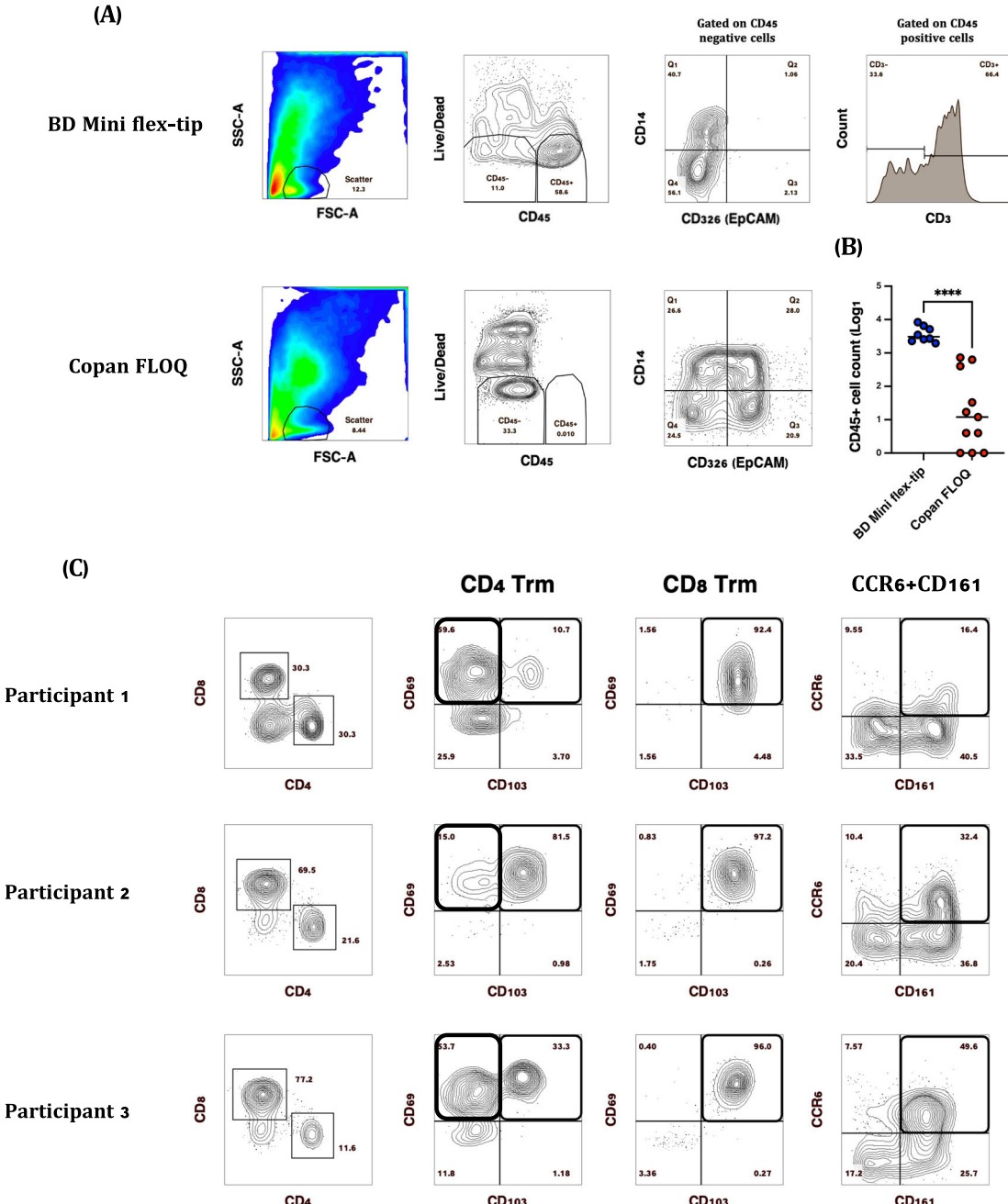

**Fig. 1 Comparison of SARS-CoV-2 diagnostic testing nasopharyngeal (NP) swabs for immune cell recovery.** Representative plots of CD45$^+$ nasal immune cells isolated from (**A**) BD Mini Flex-tip swabs, majority of which (~40–90%) were CD3$^+$ and Copan FLOQ swabs from which epithelial (CD326) (EpCAM$^+$) cells were the main cell type recovered. **B** A significant difference in CD45$^+$ immune cell recovery was observed when BD Mini Flex-tip swabs were used compared to Copan FLOQ swabs (Mann–Whitney test, ****$P > 0.0001$). **C** Representative plots of CD3$^+$ T cell subsets isolated from BD Mini-tip flexible swabs. Majority of CD4$^+$ and CD8$^+$ T cells were Trm based on the expression of CD69 and or CD103, while 16–50% of CD4$^+$ T cells were "Th17-like" based on the expression of CD161 and CCR6. was used for immune.

non-parametric Mann–Whitney test, Fig. 3). Similar trends were observed for CD40L, TNF and perforin, though these were not statistically significant. We repeated this analysis at ~6 months post vaccination, modifying the stimulation protocol, splitting samples from the same participant to examine paired antigen-specific nasal and systemic T cell responses stimulated overnight with SARS-CoV-2 spike peptides ($n = 24$). Similar to month 2 findings, stimulation with spike peptide pools resulted in significantly increased expression of CD107a (Fig. 4A) in the nasal mucosa. Deeper polyfunctional analysis indeed confirmed that a high proportion of the responding cells expressed more than one

marker, with combinations that included CD107a being the most common both in blood and the nasal mucosa. Although rare, CD8$^+$ T cells expressing 3- or 4-functions were more common in stimulated nasal samples compared to blood (Supplementary Fig. 4).

While other functions such as IFN-γ were not significant in paired analysis, we considered that not all vaccine-recipients are likely to mount antigen-specific T cell responses to SARS-CoV-2. Therefore we correlated nasal and systemic antigen-specific responses at month 6 post vaccination measured as described above, after background substraction (Fig. 5). Eight individuals

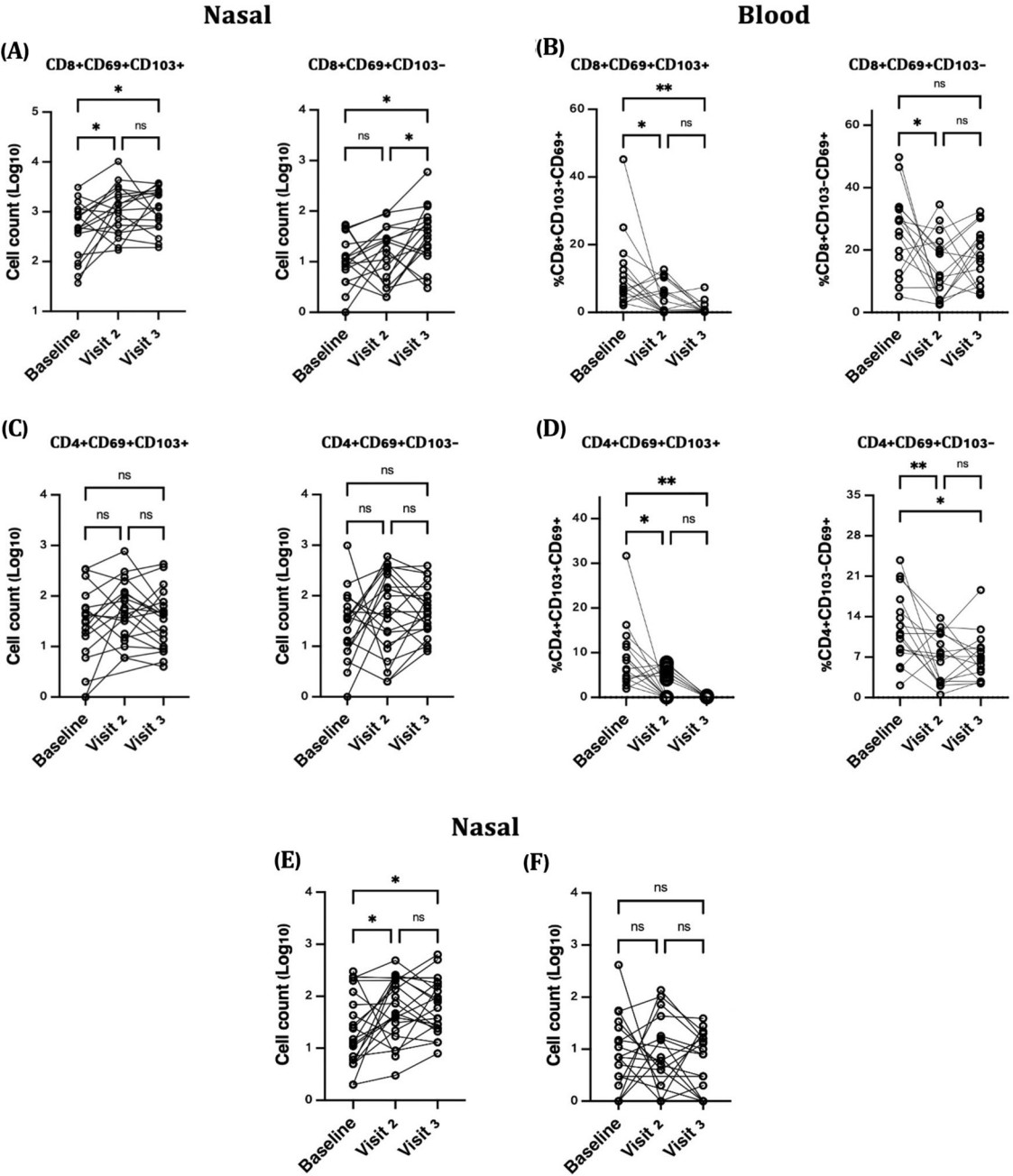

**Fig. 2 Nasal cell counts per NP swab and peripheral blood frequencies of T cell subsets post SARS-CoV-2 vaccination ($n = 20$). A** Nasal CD8+CD69+CD103+ Trm significantly increased following the first and second vaccine dose (*$P > 0.01$) while CD8+CD69+CD103− T cells increased significantly following the second vaccine dose (**$P > 0.001$). **B** No changes in nasal CD4+CD69+CD103+ Trm (ns). **C** In blood, CD8+CD69+CD103+ Trm (**$P > 0.001$) and (**D**) CD4+CD69+CD103+ decreased significantly at visit 2 and 3 compared to baseline (**$P > 0.001$). **E** Significant increases in Th17 (CD4+CD161+CCR6+) T cells at peak vaccine response time points while (*$P > 0.01$) (**F**) no changes were observed in Tfh (CD4+CXCR5+PD1+) T cells. *$P > 0.01$, **$P > 0.001$, ns non-significant.

(33%) made nasal IFN-γ responses that were >0.1% of CD8+ T cells, and these were more frequent in individuals who made vaccine responses in the blood ($p = 0.0043$, Unpaired nonparametric Mann–Whitney test). While similar connections between compartments were not observed for the CD107a and CD40L readouts, these data support the conclusion that a subset of individuals mount nasal, antigen-specific CD8+ T cell responses following SARS-CoV-2 vaccination.

To understand the relationship between antibody titers and nasal T cells, we quantified levels of SARS-CoV-2 spike-specific IgG antibodies in plasma. Antibody titers increased at visit 2

(median titer 163, IQR: 67–376), and dramatically at visit 3 (median 2185, IQR: 826–3652; Fig. 6A). Only one of two participants with prior COVID-19 was positive for SARS-CoV-2 nucleocapsid IgG (Fig. 6B). We next correlated antibody titers to the number of nasal CD8+ Trm and CCR6+ CD161+ CD4+ T cells at visit 3. Increases in both of these nasal T cells populations were similar in participants with varying increases in spike titers, with no correlations observed (Fig. 6C, D). These data suggest that tissue T cell responses to vaccination may be independent of the antibody response and may provide an additional layer of immunity against SARS-CoV-2 infection.

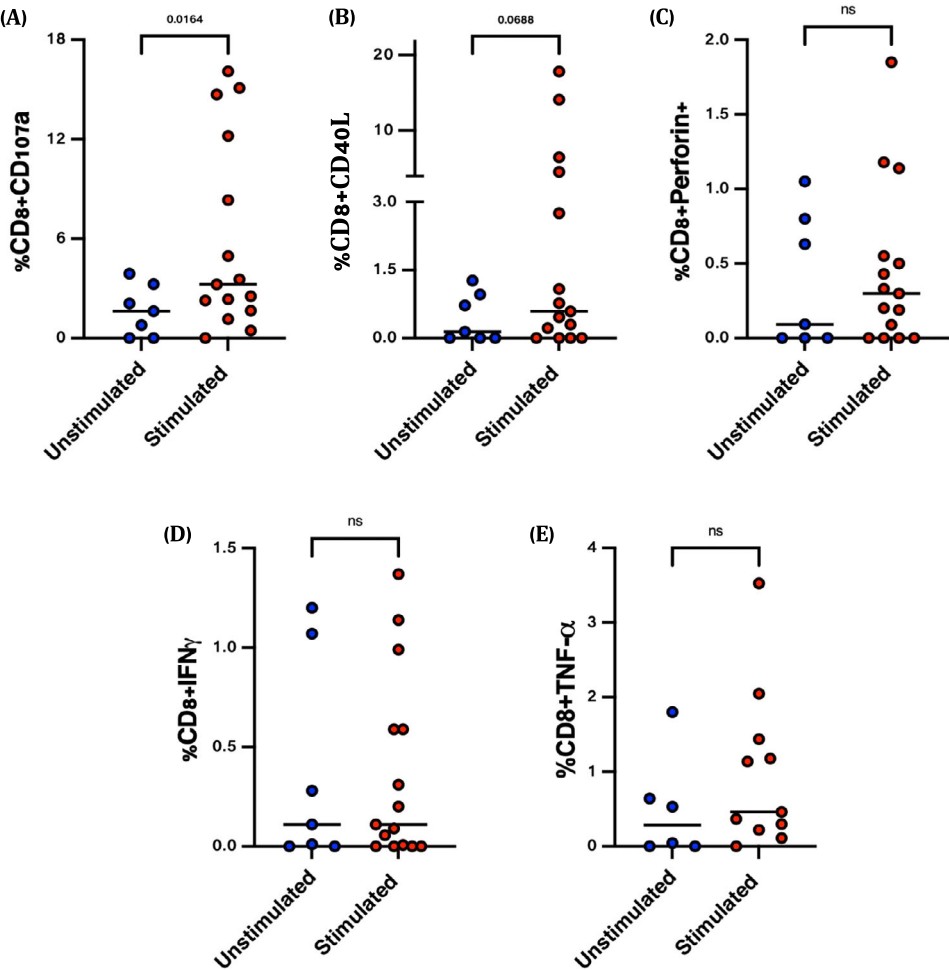

**Fig. 3 SARS-CoV-2 spike-specific CD8⁺ T cell responses ~2 months post-vaccination.** Nasal immune cells isolated from individuals, approximately 2 months post second vaccine dose, were stimulated (*n* = 15) with overlapping spike peptides (Red dots) or left unstimulated (*n* = 7) (Blue dots). Frequencies of (**A**) CD107a, (**B**) CD40L, (**C**) Perforin, (**D**) IFN-γ and (**E**) TNF antigen-specific CD8⁺ T cell responses. Unpaired non-parametric Mann–Whitney test was used (ns: non-significant).

Most immunology happens in tissues[21]. Trm cells are a vital form of surveillance in mucosal tissues, where they can mount rapid immune responses to pathogens upon re-exposure. In animal models, T cells at peak activation up-regulate homing markers that distribute these cells to relevant tissues[22], where a high proportion is retained for extended periods as a form of sentinel surveillance against re-infection[23]. Trm cells have been more difficult to study in humans, given challenges in sampling the hard-to-reach tissues and in proving residency[17,24]. Our data suggest that certain types of NP swabs commonly used for COVID-19 diagnostics are effective in recovering immune cells from nasal tissue. Multiple populations of nasal CD4⁺ and CD8⁺ T cells increased in frequency by ~0.5 orders of magnitude following vaccination with the BNT162b2 mRNA vaccine. Many of these cells were SARS-CoV-2-specific, in particular with respect to upregulation of the cytotoxicity-associated marker CD107a. This is a critical observation, as most human mucosal COVID-19 studies have focused either on antibodies[25] or lung T cells[26,27], the latter of which may not represent the front-line defense against infection.

Our data parallel comparable studies showing that COVID-19 mRNA vaccines induced mucosal IgA[28], which was subsequently shown to protect against breakthrough infection[29]. Vaccine-induced mucosal IgA was higher in convalescent individuals, suggesting that these responses could be boosted by mucosal vaccination[30]. Regarding cell-mediated immunity, it has been hypothesized that Trm cells may be the most effective form of T cell response against SARS-CoV-2 infection[31], preventing viral dissemination beyond the URT, where more virus-induced damage can occur[32]. Indeed, several recent reports suggest SARS-CoV-2 specific Trm cells can be measured either pre-infection[33] or in convalescent individuals[34–36], suggesting that exposure via the natural route may augment nasal Trm cell development. Regarding vaccination, data from mice have suggested that parenteral vaccination induces nasal Th17 Trm cells that protect against bacterial colonization[37,38]. The findings from our study however suggest that nasal T cells may be induced by parenteral vaccination against COVID-19.

Our study had some limitations. We enrolled a relatively small sample size, restricting our ability to extensively stratify our data. The age range does not include the elderly, a group most in need of protection by COVID-19 vaccination[39]. The immune cell recovery from NP swabs, while sufficient for phenotyping and enumeration, is challenging for extensive functional characterization. We are currently assessing alternative approaches to expand this aspect of the analysis. However, the major advantage provided by minimally-invasive, longitudinal sampling may supersede these concerns, in particular for certain research questions that can be addressed in these types of samples.

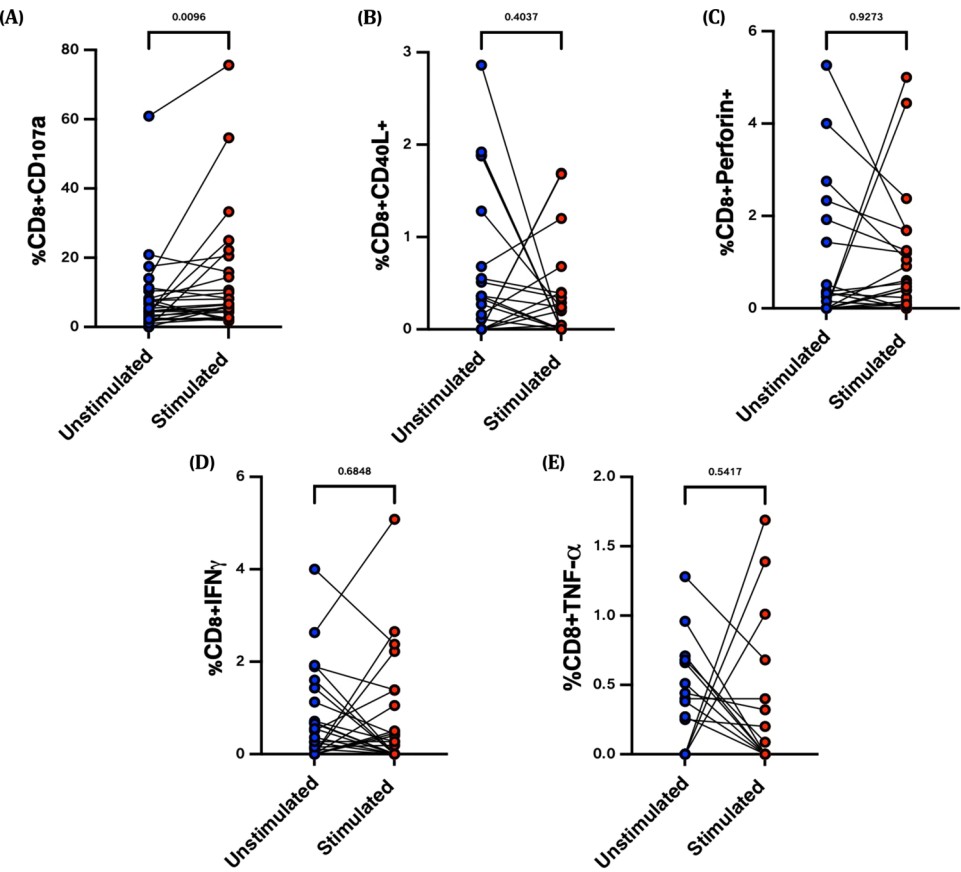

**Fig. 4 SARS-CoV-2 spike-specific CD8+ T cell responses ~6 months post-vaccination.** Nasal immune cells isolated from ($n = 24$) participants, were stimulated with overlapping spike-specific peptides (Red dots) or left unstimulated (Blue dots) for ~16 h. The figures show the frequencies of (**A**) CD107a, (**B**) CD40L, (**C**) Perforin, (**D**) IFN-γ and (**E**) TNF antigen-specific CD8+ T cell responses. Wilcoxon matched-pairs signed rank test was used.

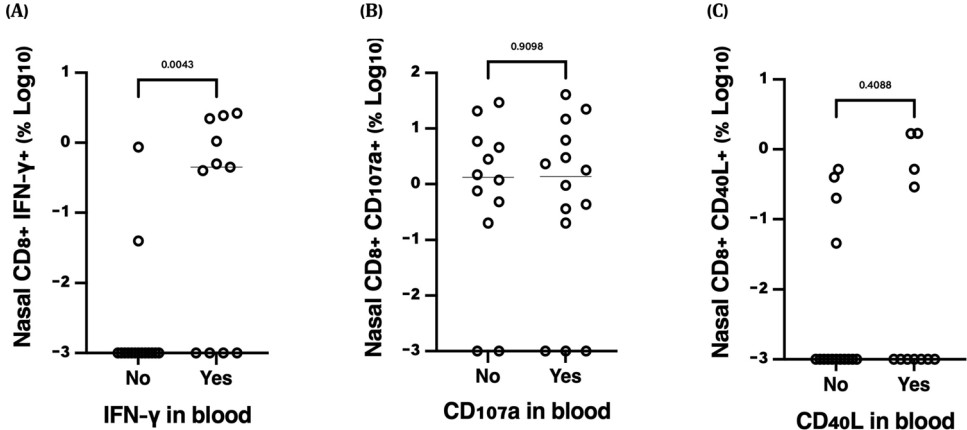

**Fig. 5 Nasal-blood correlations between background-subtracted SARS-CoV-2 spike-specific CD8+ T cell responses ~6 months post-vaccination.** Nasal immune cells isolated from ($n = 24$) participants, were stimulated with overlapping spike-specific peptides or left unstimulated for ~16 h. Nasal CD8 + T cell responses for (**A**) IFN-γ, (**B**) CD107a, and (**C**) CD40L are shown, stratified by whether a positive response for that readout was detected in matched blood. Blood responses below detected were imputed as "0.001" for graphing purposes. Unpaired non-parametric Mann–Whitney test was used.

In summary, nasal T cell studies are feasible in humans, but this is highly dependent on the type of NP swab that is used. A high proportion of nasal T cells express markers of tissue residency, and these cells increase significantly in number and function following SARS-CoV-2 mRNA immunization. Several recent data, mainly in animal models, suggest that heterologous vaccination that combines a mucosal and systemic route offer more robust protection against infection, supporting the case for a more potent, transmission-blocking vaccine[40–45]. This is particularly pertinent given the recent omicron variant wave, where a higher rate of SARS-CoV-2 breakthroughs have been observed[46,47]. Further work should be performed in humans to understand the role that local T cells may play in the protection against SARS-CoV-2 infection.

## Methods

**Study participants and ethics board approval.** Volunteers ($n = 29$) for this observational clinical study included staff enrolled at the Cadham Provincial

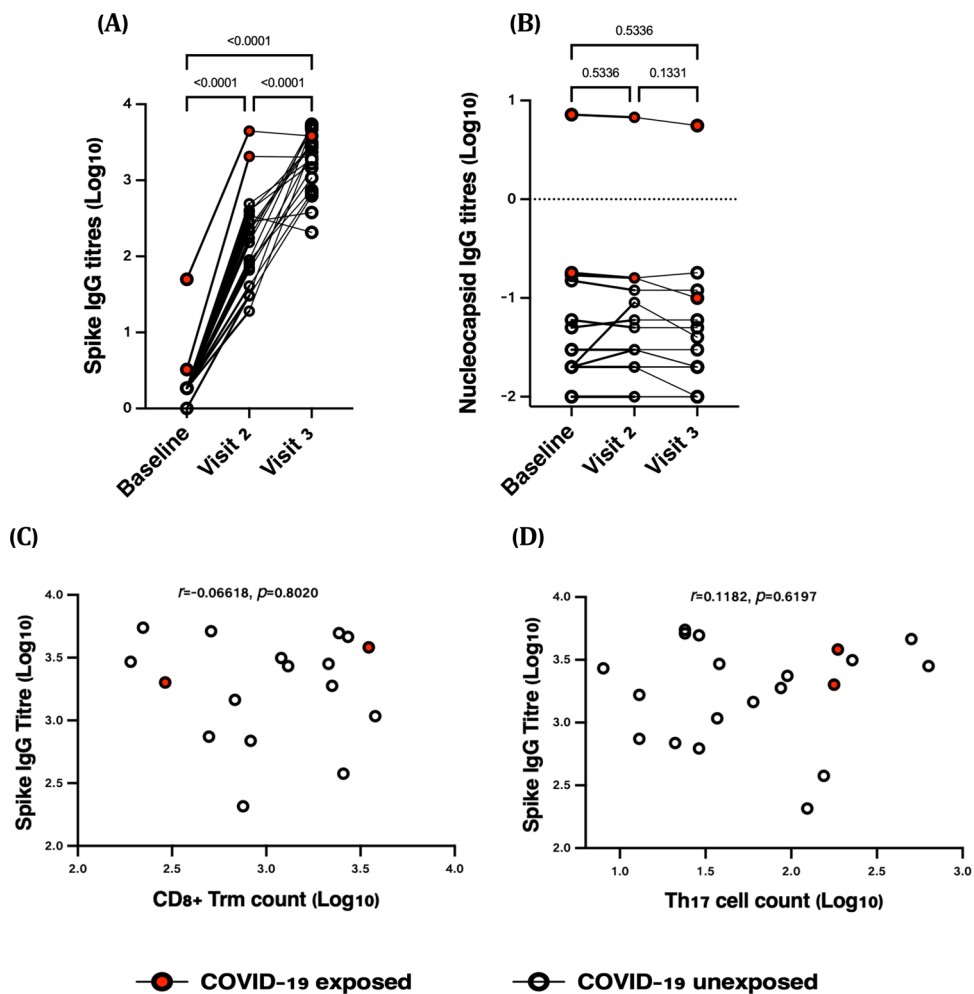

**Fig. 6 Plasma IgG antibody titers following SARS-CoV-2 vaccination and correlations with nasal T cell responses ($n = 21$). A** Spike-specific IgG titers increased significantly from baseline, visit 2 and visit 3 ($P < 0.0001$) while **B** no differences were observed for nucleocapsid specific IgG titers. **C** Correlation of nasal CD8 + Trm and **D** CD4$^+$ CCR6$^+$CD161$^+$(Th17) T cell counts with spike antibody titers at visit 3, ~12 days following the second SARS-CoV-2 vaccine dose. Red dots indicate COVID-19 exposed participants while the white dots represent COVID-19 unexposed participants (Two-tailed non-parametric Spearman's correlations test).

Laboratory in Winnipeg. Ethical approvals were obtained from the University of Manitoba Health Research Ethics Board (HREB) and the Public Health Agency of Canada (PHAC). Informed consent was obtained and a comprehensive socio-demographic and health assessment questionnaire administered to all study participants.

**Sample collection and processing**. In pilot work, we compared the BD Flex-mini and Copan FLOQ swabs. In the vaccine study participants, matched diagnostic pharyngeal (NP) swabs (BD Flex-mini) and peripheral blood samples (BD EDTA Vacutainer) were collected prior to vaccination (baseline), ~12 days after the first vaccine dose (visit 2), and ~12 days after the second dose (visit 3). Nasal cells were isolated from NP swabs which were inserted into the nose, rotated once (360º) and placed into a tube containing 3 ml of viral transport medium (VTM) and placed on ice for processing within 2 hours of collection. Once in the laboratory, swabs were vigorously vortexed to dissociate the cells and mucus and centrifuged for 10 min at 1600rpm. NP swabs were then rinsed with PBS 2% FBS, and the cell suspension filtered through 100 µm nylon cell strainer (Becton Dickinson) fitted into a 50 ml tube. Cells were washed twice prior to flow cytometry staining. Peripheral blood mononuclear cells (PBMC) were isolated by ficoll density gradient centrifugation using SepMate 50 (STEMCELL technologies) following the manufacturer's instructions.

**Flow cytometric staining of nasal cells**. Cells were stained with viability exclusion dye, together with a panel of pre-titrated surface antibodies (Supplementary Table 1) in PBS 2% FBS on ice for 30 min. The NP swab optimization experiment (Fig. 1) included anti-human CD326 (EpCAM) (Biolegend, Clone 9C4) in the surface staining cocktail to stain for epithelial cells. Cells were subsequently washed, permeabilized in fixation/permeabilization solution (eBiosciences) and

stained for Ki-67 and perforin. The cells were washed again and resuspended in perm/wash buffer (eBioscience). Data were acquired using a BDLSR Fortessa flow cytometer (BD Biosciences) and analyzed using FlowJo version 10.7.1 (Becton Dickinson Life Sciences).

**Antigen-specific stimulation and flow cytometric staining of nasal and per-ipheral blood mononuclear cells**. Approximately 6 months post-2nd vaccination, freshly isolated nasal cells and peripheral blood mononuclear cells (PBMCs) from ($n = 24$) were stimulated for 16 hours at 37 °C / 5% $CO_2$ with a SARS-CoV-2 peptide pool (PepTivator, Miltenyi Biotec), comprised of 15-mer sequences over-lapping by 11 amino acids, encompassing the complete protein coding sequence (aa 5–1273) of the spike glycoprotein ("S") of SARS Coronavirus 2 (GenBank MN908947.3, Protein QHD43416.1) at a final concentration of 1 µg/mL, in the presence of Golgi-Plug (BD Biosciences), Golgi-Stop (BD Biosciences) and anti–CD107a (clone H4A3, BD Biosciences). As a control for this experiment, matched cells were left unstimulated. Following stimulation, cells were washed, stained with Live/Dead Fixable Aqua viability dye (Invitrogen) for 30 min on ice. Cells were subsequently washed, permeabilized with fixation/permeabilization solution (eBiosciences) and stained in a cocktail of pre-titrated monoclonal anti-bodies (Supplementary Table 1). Cells were then washed, acquired using a BDLSR Fortessa (BD Biosciences) and analyzed using FlowJo version 10.7.1 (Becton Dickinson Life Sciences) and SPICE (Simplified Presentation of Incredibly Com-plex Evaluations)[48]. Antigen specific responses were determined by background subtraction of the stimulated cells from the unstimulated cells.

**Plasma antibody testing**. Concentrations of SARS-CoV-2-specific IgG were determined using commercial assays. At visit 3, plasma was diluted 1:10 because of the high magnitude of the responses. IgG recognizing the SARS-CoV-2 spike

protein was quantified using the Diasorin Trimeric Spike ELISA, while nucleocapsid antibodies were measured using the ELISA from Abbott.

**Dimensionality reduction, visualization and clustering**. FlowJo version 10.7.1 (Becton Dickinson Life Sciences), was utilized to dimensionally reduce and to interrogate immune cell clusters and phenotypes in cells isolated from nasal swabs. Briefly, manual gating was performed to exclude debris, doublets and dead cells. Live CD45+ cells from all participants ($n = 21$) at baseline, visit 2, and visit 3, were integrated into a single file which was dimensionally reduced and visualized using t-distributed stochastic neighbor embedding (tSNE), following the default parameter settings. FlowSOM algorithm (Version 2.6) was used to create cluster populations from the dimensionally reduced data based on their similarity and subsequently segregated based on relative abundance and phenotype using ClusterExplorer (Version 1.5.9).

**Statistics**. Clinical and demographic data are presented as proportions or medians and interquartile ranges. Linear mixed models were used to compare changes in log10 transformed immune cell abundance (in nasal mucosa, typically cells/swab; in blood, % of CD3+CD4+ or CD8+ T cells). In these models, visit as a categorical variable was the main predictor, with visit 2 and 3 compared to baseline. Multivariate models were adjusted for a range of covariates. Raw p values, beta coefficient and 95% confidence intervals were reported. Statistical analyses were performed using SPSS v. 27.0 while all the graphing and was performed using Prism version 9 (GraphPad Software, LLC).

**Reporting Summary**. Further information on research design is available in the Nature Research Reporting Summary linked to this article.

## Data availability

All data are available in the main article and its supplementary files or from the corresponding author upon reasonable request. Source data are provided with this paper.

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

## Acknowledgements
This work was funded by the Bill and Melinda Gates Foundation (INV-032575) and the Public Health Agency of Canada (PHAC). Special thanks to the study volunteers whose time and willingness to provide repeated specimens made this project possible.

## Author contributions
A.S,. D.S,. L.R.M.: Conceptualized the study, designed the experiments and analysed data. L.R.M., T.B.B., J.B., P.V.C.: Overall study supervision and coordination. C.M.C., S.K., Y.K,. R.S., H.J., P.J.M,. T.BB.,. J.B., P.V.C., D.S.: Provided intellectual input into study design. N.J., G.S., B.A.: Enrolled study participants, carried out sampling and/or collected and entered data. A.S., H.M.N., F.N., D.S.: Performed experiments. A.S., L.R.M.: Wrote the initial draft manuscript. All authors reviewed and approved the submitted manuscript.

## Competing interests
The authors declare no competing interests.
