## [Peer Review File · Nature Communications]

REVIEWER COMMENTS

Reviewer #1 (Remarks to the Author):

In the manuscript the authors investigate the proportions and numbers of nasal CD8 Trm cells after immunisation in human volunteers with an RNA vaccine. They observed that the number of CD69+/CD103+ CD8 cells increased after immunisation. There was a modest increase in CD107a after stimulation, but no other marker. CD8 counts did not correlate with antibody responses to the vaccine.

Other papers to consider: Exposito et al <https://pubmed.ncbi.nlm.nih.gov/34021148/> Nguyen <https://pubmed.ncbi.nlm.nih.gov/33532071/> (both of which look at Trm after infection) but a useful comparator.

1. Experimental:

- a. How were the cells counted after recovery? The % of CD8 (supplementary figure 1) is unchanged overtime, so the increase represents increased numbers of cells. How confident can the authors be that this is not a recovery or sampling issue.
- b. There is no strong evidence that the cells are Spike specific. The only indication is an increase in CD107 after stimulation, there is no control peptide. Some other method of interrogation e.g. ELISPOT might have been helpful.
- c. The description as TH17 cells is based on cell surface markers alone, it might be more appropriate to describe the cells by their surface markers not assign a functional type.
- d. What does the linear mixed model add?
- e. How were the lymphocytes gated? Did you use a back gate from CD3 – there is a lot of cells not being considered.
- f. Curious that there is 3 clear populations in the live dead gate for coplan (fig 1A)

2. Impact: how do the findings advance the field, either of vaccines or general mucosal immune responses?

- a. It is a useful demonstration that these cells can be recovered from nasal tissue with a nasal swab, but this has been demonstrated elsewhere.
- b. It is a small study (as admitted by the authors) but this does detract from the impact.
- c. The lack of relationship between CD8 and antibody does not seem overly surprising

Minor

1. The abstract needs a bit more of the key methods, would help to say it is in human volunteers in abstract or title.
2. CD154 is not significant ($p > 0.05$)
3. State how many donors in the figure legends

Reviewer #2 (Remarks to the Author):

The manuscript "Expansion of tissue-resident CD8+ T cells and CD4+ Th17 cells in the nasal mucosa following mRNA COVID-19 vaccination" by McKinnon and colleagues characterizes T cell populations in 21 healthy individuals before and after vaccination with the Pfizer-BioNTech vaccine. The study first establishes that certain swabs are much better than others at retrieving T cells from the nasal mucosa. Secondly, most of the CD8+ T cell were CD69 and CD103 positive (markers associated with resident memory T cells) and rose in frequency and number following vaccination while those in the blood were reduced. They also show evidence of cytotoxic activity and cytokine production. Third, CD4+ T cells were stable except for a noted increase in Th17 cells. Interestingly, there was no association between spike IgG antibody titers and T cells in the nasal mucosa. The methodology, data analysis, and interpretation presented are sound and the authors have highlighted the limitations of the study.

Overall, this work provides important information on an improved methodology to track T cell responses in the nasal mucosa and, although the sample size is small, the investigators begin to dissect characteristics of cell-mediated immunity within a relevant mucosal target tissue following muscular injection of a COVID-19 vaccine. Thus, I think the information is novel and timely. Comments below relate to improving presentation and discussion of the data.

1. Unlike CD8+ T cells, CD4+ TRM expression of CD103 is variable (PMID: 28930685). I think this should be commented on from the beginning related to Figure 1C- meaning the CD4+CD69+CD103- population could also be TRM.
2. Figure 2B in the text describes CD4+ T cells but the data is about CD8+ T cells.
3. Figure 2B- percentages are from what population? Percentage of CD3+ cells? It's surprising there are that many CD69+ cells in the blood.
4. Figure 3- could the data be presented as single, double, triple, etc. expressing cells? Speaks to the level of poly-functionality of the CD8 pool. Also, was this assay performed on blood samples, and if so, is there any difference?
5. Figure 4- could the data points for the two people who had prior COVID-19 exposure be a different shape or color? Although it's only two data points, it's still of interest to see where those samples fall within the data range.
6. Supplemental Figure 2- it looks like there are significant differences in HLA-DR, CD38, perforin, etc. but the text says there are no changes.

Minor:

1. Suggestion to add 'Th17' and 'Tfh' headings to Figure 2E and 2F to make it easier for the reader.

Reviewer #3 (Remarks to the Author):

The manuscript by Seemaganda et al., investigates the changes in immune cell composition in the nasal mucosal following COVID-19 vaccination. The authors show that vaccination with the BioNTech mRNA vaccine results in the increase in the number of tissue resident memory T cells in the nasal tissue. This is a unique study as unlike other reports assessing vaccine induce T cell responses in peripheral blood, this report focuses on the impact of vaccination on the immune cell subsets in the nasal mucosa – clearly an extremely important site for infection control. While the report is significant, the data presented is light and several of the conclusions are not well supported by the presented data and thus further experimentation is required.

1. Figure 1 - To support the validity of using these BD swabs for T cell analysis of the nasal mucosa it is important to show how variable the yield in cell recovery is across patients and time points. I presume the authors already have this information. It is important to graphically show the spread in total cell recovery. Moreover, it is important to show that this testing approach can yield consistent results. It would be interesting to see if the authors sampled the same healthy donor on two separate occasions whether the immune cell profile in the nasal tissue would be consistent.
2. Figure 1C – It should be noted that while CD103 and CD69 are often used as gold standard markers to define CD8+ Trm it is well accepted that CD4+ Trm do not necessarily express CD103. I recommend the authors consider using CD69+ as a marker for CD4 Trm

3. Figure 1C – I am not completely convinced by the use of CCR6 and CD161 alone to define Th17 CD4 T cells. I recommend the authors stimulate these cells with PMA/ION and show the cytokine profile. Alternatively, a genetic analysis (RNAseq) of nasal vs PBMC CD4 T cells might help support the authors conclusion that these cell are truly Th17. Also, can the authors explain the rationale for specifically looking for Th17 CD4 Trm – is there a rationale as to why these would be more protective against COVID-19?
4. Figure 2b – CD4 Trm should be defined as CD69 +/- CD103 (not the other way around)
5. Figure 2b – I am confused by the analysis of Trm markers on T cells in the blood. The authors propose that the drop in cells with Trm markers in the blood correlates with an increase in Trm in the nasal tissue and thus speculate that this is because these cells are being recruited from the blood into the nasal tissue. Trm cells do not re-circulate and to my knowledge are not thought to be recruited from a pool of cells in the blood which express these phenotypic markers. Rather Trm cells are thought to develop in situ in the tissue. I strongly recommend the authors rethink this analysis and interpretation of their data. Perhaps alternative analysis on blood t cells (activation makers, memory markers, responsiveness to peptide stimulation etc) might be more useful
6. Supplementary Fig 2 – it is strange that no activation markers are observed on T cells in the blood or nasal tissue following vaccination. Can the authors comment on why they think this is occurring? Moreover, can they confirm that in the blood they were gating on antigen experienced (CD45RO+) cells?
7. Supplementary Figure 3 – Further work should be done to define the cell type the CD3-CD4-CD8- cells and the CD3+CD4-CD8- pool represent – these cells seem important as they are expanding following vaccination? RNAseq analysis would be useful here as cell numbers are low.
8. Figure 3 – Can a control be performed stimulating T cells recovered from the baseline sample of these donors. Or if limiting, using T cells recovered from unvaccinated individuals would also be useful to show that the observed response is specific. Moreover, can the authors show whether activation markers profiled earlier (ie HLA-DR and CD38) become elevated following stimulation? In addition, it would be useful to show a similar analysis on the matched blood samples.
9. Figure 4 – Although total CD8 Trm numbers in the nasal tissue do not correlate with Ab titres perhaps antigen specific CD8 Trm do? It would be interesting to see the % of CD8+TNF+ or CD8+IFNg+ cells post stimulation plotted against Ab titre.

Minor

- Fig 2 – part F labelled incorrectly in the figure legend

REVIEWER COMMENTS

Reviewer #1 (Remarks to the Author):

In the manuscript the authors investigate the proportions and numbers of nasal CD8 Trm cells after immunisation in human volunteers with an RNA vaccine. They observed that the number of CD69+/CD103+ CD8 cells increased after immunisation. There was a modest increase in CD107a after stimulation, but no other marker. CD8 counts did not correlate with antibody responses to the vaccine.

Other papers to consider: Exposito et

al <https://pubmed.ncbi.nlm.nih.gov/34021148/> Nguyen <https://pubmed.ncbi.nlm.nih.gov/33532071/> (both of which look at Trm after infection) but a useful comparator.

We thank the reviewers for these references, which are now cited in the revised manuscript.

1. Experimental:

a. How were the cells counted after recovery? The % of CD8 (supplementary figure 1) is unchanged overtime, so the increase represents increased numbers of cells. How confident can the authors be that this is not a recovery or sampling issue.

This is an important point raised by the reviewer. In all of our mucosal studies we typically acquire the entire sample and use the absolute number of cells as an outcome in analyses. This is because capturing absolute number is often more critical in mucosae as it captures the influx of cells that may occur as a result of a biological stimulus. While the frequency of marker expression on cells may differ by a small proportion, the total number of cells often differs by orders of magnitude, particularly in inflamed versus uninfamed context. Because we anticipated that vaccination may similarly introduce a population of antigen-specific T cells, we hypothesized that the absolute number of cells per swab may be an important outcome in this study, which we feel is indeed reflected by the data. One final clarifying point -- it is also important to note that the frequency (%) CD8+ Trm is not “unchanged” – the frequency increases after vaccination, but it is quite high to start with and perhaps that explains the lack of statistical significance.

b. There is no strong evidence that the cells are Spike specific. The only indication is an increase in CD107 after stimulation, there is no control peptide. Some other method of interrogation e.g. ELISPOT might have been helpful.

We thank the reviewer for this comment and have addressed it by generating new data, presented in revised Figures 4 and 5. We did not run ELISPOT assays as these tend to capture only one or a few functions of T cells, even though it does have a high sensitivity. This is something that could be considered in follow-up studies. In the original submission, at 2-months post-vaccinations, we randomized vaccine recipients to have their cells stimulated with spike peptides or left in media for 4 hours, therefore compared unstimulated to stimulated cells (in an unpaired fashion) instead of control peptide. We felt this was a reasonable control, as we don't know what peptides the cells at this site would not respond to for certain. The reason for the short incubation was in case cell viability would dramatically decrease during overnight stimulation. We have since tested this and found a high viability following overnight culture. Therefore, to better define the antigen-specificity of nasal T cells post-vaccination, in the new submission we have added data using paired Spike-stimulated versus unstimulated overnight intracellular cytokine assays from the same individual. These data confirm that CD107a increases in a SARS-CoV2 Spike-specific fashion at 6 months post-vaccination in most participants. There are also examples of responses above background for the other readouts,

but as is the case in the blood, not every participant makes T cell responses to this vaccine regimen. For example, nasal CD8+ IFN γ responses are significantly above background (unstimulated cells) in 8/24 participants (comparable to 11/24 in blood). Either way, the strongest signal of a Spike-specific response to our data is CD107a, suggesting that cytotoxicity may be an important phenotype of this subset (now reflected in our revised title).

c. The description as TH17 cells is based on cell surface markers alone, it might be more appropriate to describe the cells by their surface markers not assign a functional type.

This is a valid point. We have changed the description of these cells as “CCR6+ CD161+” in the revised title and text, and comment that these markers have been used in the past to signify Th17 (but are only putative markers).

d. What does the linear mixed model add?

We chose this analysis method as it is most appropriate to analyze repeated measures data while also controlling for covariates that may be associated with either predictor or outcome. Importantly, a paired T-test gave similar results for unadjusted outcomes, with the limitation that covariates cannot be taken into account.

e. How were the lymphocytes gated? Did you use a back gate from CD3 – there is a lot of cells not being considered.

Lymphocytes were gated based on forward versus side scatter. To investigate the reviewer’s CD3 question, we back-gated the CD3+ population to determine where these cells were distributed on CD45+ versus live dead, singlets, and forward versus side scatter (see figure below). This analysis confirms that all the cells that were gated out were either dead or CD3 negative cells. In the representative figure below, we included a majority of the cells in the scatter plot, and back gating on the CD3+ population suggests that most of these cells are CD45 negative and/or dead.

Curious that there are 3 clear populations in the live dead gate for coplan (fig 1A)

Although we can’t be completely sure of the reason for this, three CD45 negative populations were typical for nasal cells in a majority of participants. These could be different populations of non-immune cells, such as fibroblasts or others; scRNAseq experiments of BAL suggest that many types of immune cells such as mDCs, pDCs, mast cells, NK cells, T cells and B cells are observed (<https://doi.org/10.1038/s41591-020-0901-9>). Since the live/dead gate was set based on fluorescence minus one control, we are confident from multiple experiments that the live-dead+ populations are likely to be two different populations of dead/dying cells and not the cells of interest for this analysis.

2. Impact: how do the findings advance the field, either of vaccines or general mucosal immune responses?

a. It is a useful demonstration that these cells can be recovered from nasal tissue with a nasal swab, but this has been demonstrated elsewhere.

We appreciate the positive feedback. We also note that there are very limited data available for characterizing T cells in the upper respiratory tract (URT) of humans, even outside of the context of vaccination (COVID or otherwise). Many mucosal studies of COVID in humans either focus on mucosal antibodies (highly relevant, but different than T cell responses), or use RNAseq approaches from cells collected deeper in the lung (e.g. by bronchoalveolar lavage). Others use tissues collected during surgery, which is not an ideal sample type for routine monitoring in longitudinal studies. Therefore, we feel our study provides novel insights into COVID vaccine immunogenicity but also a strong rationale for future longitudinal nasal studies in either vaccine recipients or individuals infected by respiratory pathogens. We have added a comment on this in the revised Discussion and updated the references to highlight recent publications and pre-prints that have emerged since our original submission.

b. It is a small study (as admitted by the authors) but this does detract from the impact.

c. The lack of relationship between CD8 and antibody does not seem overly surprising

Minor

1. The abstract needs a bit more of the key methods, would help to say it is in human volunteers in abstract or title.
2. CD154 is not significant ($p > 0.05$)
3. State how many donors in the figure legends

We have made these minor changes in the revised manuscript.

Reviewer #2 (Remarks to the Author):

The manuscript "Expansion of tissue-resident CD8+ T cells and CD4+ Th17 cells in the nasal mucosa following mRNA COVID-19 vaccination" by McKinnon and colleagues characterizes T cell populations in 21 healthy individuals before and after vaccination with the Pfizer-BioNTech vaccine. The study first establishes that certain swabs are much better than others at retrieving T cells from the nasal mucosa. Secondly, most of the CD8+ T cell were CD69 and CD103 positive (markers associated with resident memory T cells) and rose in frequency and number following vaccination while those in the blood were reduced. They also show evidence of cytotoxic activity and cytokine production. Third, CD4+ T cells were stable except for a noted increase in Th17 cells. Interestingly, there was no association between spike IgG antibody titers and T cells in the nasal mucosa. The methodology, data analysis, and interpretation presented are sound and the authors have highlighted the limitations of the study.

Overall, this work provides important information on an improved methodology to track T cell responses in the nasal mucosa and, although the sample size is small, the investigators begin to dissect characteristics of cell-mediated immunity within a relevant mucosal target tissue following muscular injection of a COVID-19 vaccine. Thus, I think the information is novel and timely. Comments below relate to improving presentation and discussion of the data.

We thank the reviewer for these positive comments.

1. Unlike CD8+ T cells, CD4+ TRM expression of CD103 is variable (PMID: 28930685). I think this should be commented on from the beginning related to Figure 1C- meaning the CD4+CD69+CD103- population could also be TRM.

We have made this change in the revised manuscript.

2. Figure 2B in the text describes CD4+ T cells but the data is about CD8+ T cells.

We apologize for this error and have made the necessary correction.

3. Figure 2B- percentages are from what population? Percentage of CD3+ cells? It's surprising there are that many CD69+ cells in the blood.

The proportions displayed are a percentage of CD3+CD8+ T cells. While CD69 expression in blood is typically low, we have found previously that a median of ~20% of CD4+ T cells in HIV uninfected Kenyan women can express CD69 (McKinnon et al *J Immunol* 2011). This is similar to what we observe in this vaccine cohort. These could represent recently activated cells, or possibly ex-TRM. While a full explanation regarding the presence of these cells post-vaccination is unclear, we feel that their decline in blood coupled with increase in nasal tissue is worth reporting, given the scope of the current work.

4. Figure 3- could the data be presented as single, double, triple, etc. expressing cells? Speaks to the level of poly-functionality of the CD8 pool. Also, was this assay performed on blood samples, and if so, is there any difference?

We thank the reviewer for this suggestion. We have added polyfunctionality analyses using SPICE to the new data on nasal T cell at 6 months post-vaccination (revised Supplemental Figure 4, same data as revised Figures 4 & 5). To measure these responses, we divided nasal cells into 2 vials and compared media versus complete SARS-CoV-2 Spike peptide pool (Miltenyi-Biotec) stimulated for 16 hours. Matched blood samples were also assessed in this assay at the same time point. The outer arcs highlight that CD107a is often part of nasal CD8 responses, even when more than one readout is positive in the same cell.

5. Figure 4- could the data points for the two people who had prior COVID-19 exposure be a different shape or color? Although it's only two data points, it's still of interest to see where those samples fall within the data range.

This is a good suggestion. We have now indicated these 2 participants in the revised graphs.

6. Supplemental Figure 2- it looks like there are significant differences in HLA-DR, CD38, perforin, etc. but the text says there are no changes.

We have now ensured that the text and figures both represent the accurate statistically significant trends in the data.

Minor:

1. Suggestion to add 'Th17' and 'Tfh' headings to Figure 2E and 2F to make it easier for the reader.

We have made this change.

Reviewer #3 (Remarks to the Author):

The manuscript by Ssemaganda et al., investigates the changes in immune cell composition in the nasal mucosal following COVID-19 vaccination. The authors show that vaccination with the BioNTech mRNA vaccine results in the increase in the number of tissue resident memory T cells in the nasal tissue. This is a unique study as unlike other reports assessing vaccine induce T cell responses in

peripheral blood, this report focuses on the impact of vaccination on the immune cell subsets in the nasal mucosa – clearly an extremely important site for infection control. While the report is significant, the data presented is light and several of the conclusions are not well supported by the presented data and thus further experimentation is required.

We thank the reviewer for their positive comments and have made several of the requested changes as indicated below.

1. Figure 1 - To support the validity of using these BD swabs for T cell analysis of the nasal mucosa it is important to show how variable the yield in cell recovery is across patients and time points. I presume the authors already have this information. It is important to graphically show the spread in total cell recovery. Moreover, it is important to show that this testing approach can yield consistent results. It would be interesting to see if the authors sampled the same healthy donor on two separate occasions whether the immune cell profile in the nasal tissue would be consistent.

We are happy to clarify this point for the reviewer. The spread of cell recovery is indeed a primary outcome of the analyses, as the absolute number of cells recovered per swab is an important measure of mucosal tissue T cell influx, which we hypothesized might be induced by the vaccine. Several of the graphs in the manuscript display the number of cells recovered, including in pilot optimization experiments, so the distributions can be readily assessed. While we appreciate the idea of sampling the same healthy donor over time, it is important to point out that there could be true biological variability between time points that is not purely technical (e.g. could be influenced by environmental exposures at the time of sampling). Also, biological replicates are not possible as a second swab at the same visit may naturally provide different results than the first (e.g. if swab #1 disturbs the mucosal barrier).

To overcome this limitation in the kinetics study (Figure 2), we carried out a paired analysis pre- and post-vaccine, such that each individual served as their own control. Furthermore, while we can't link the same individuals, we found a very similar distribution of the # of cells recovered in pilot work compared to the baseline pre-vaccine visit, with most participants having between 10^3 and 10^4 CD45+ cells/swab. Unfortunately, we are not able to link individuals from the pilot and main study due to the fact that the pilot samples did not have a study ID. Despite this limitation, we feel that this sampling method offers a robust way to enumerate nasal T cells that was consistent across multiple experiments, and to determine how the numbers of different subsets changes after vaccination.

2. Figure 1C – It should be noted that while CD103 and CD69 are often used as gold standard markers to define CD8+ Trm it is well accepted that CD4+ Trm do not necessarily express CD103. I recommend the authors consider using CD69+ as a marker for CD4 Trm

We agree with this point, which was also raised by R2. We have now referred to CD69+ CD103+/- CD4+ T cells as both being Trm.

3. Figure 1C – I am not completely convinced by the use of CCR6 and CD161 alone to define Th17 CD4 T cells. I recommend the authors stimulate these cells with PMA/ION and show the cytokine profile. Alternatively, a genetic analysis (RNAseq) of nasal vs PBMC CD4 T cells might help support the authors conclusion that these cells are truly Th17. Also, can the authors explain the rationale for specifically looking for Th17 CD4 Trm – is there a rationale as to why these would be more protective against COVID-19?

We agree with the point about CCR6 and CD161 possibly indicating 'Th17-like' rather than *bona fide* Th17 cells, and as indicated in response to a similar comment from R1, we now refer to this subset by its surface markers. We also appreciate the suggestion to use PMA/IO and have done this in prior studies; however, we are unable to go back to stored nasal cells (all are

consumed in the present analysis) and furthermore anticipate there may be technical challenges with such a potent stimulus in a mucosal sample with modest cell recovery. Rather than a polyclonal stimulus, we instead opted to focus the limited cells recovered toward defining antigen-specificity of T cells, with the results presented in Figures 4 and 5. Similarly, RNA-seq would need to be optimized for this sample type and will be the focus of future studies.

Regarding the rationale for measuring Th17 (or Th17-like) cells, this is an important mucosal cell population that has been induced by previous vaccine studies (see: PMID 31659300, 30127384), and one that may be critical for maintain mucosal barriers that could protect against other infections.

4. Figure 2b – CD4 Trm should be defined as CD69 +/- CD103 (not the other way around)

We agree with this suggestion, as indicated above, and have now made this change.

5. Figure 2b – I am confused by the analysis of Trm markers on T cells in the blood. The authors propose that the drop in cells with Trm markers in the blood correlates with an increase in Trm in the nasal tissue and thus speculate that this is because these cells are being recruited from the blood into the nasal tissue. Trm cells do not re-circulate and to my knowledge are not thought to be recruited from a pool of cells in the blood which express these phenotypic markers. Rather Trm cells are thought to develop in situ in the tissue. I strongly recommend the authors rethink this analysis and interpretation of their data. Perhaps alternative analysis on blood t cells (activation makers, memory markers, responsiveness to peptide stimulation etc) might be more useful

The reviewer makes a good point about Trm being induced in tissue. However, several reports have suggested that CD69 and CD103 can be expressed in the blood (e.g. PMID 33115867), and (for multiple reasons) we do observe significant decreases in these (and other activation markers, see Suppl. Fig. 2A) post-vaccination. While the reasons for this decrease are unclear, we did feel that it was important to carry out paired blood and nasal T cell analyses. It should also be pointed out CD8+ Trm have been found to recirculate, at least in mice, with preferential mucosal homing retained in “ex-Trm” (Fonseca et al *Nat Immunol* 2020, PMID: 32066954). While the extent to which this happens in humans is unknown, we feel that presentation of blood CD69 and CD103 expression could be of interest for future studies. We have now refrained from referring to this blood population as “Trm” and have tempered the discussion around this topic in the revised draft.

Further to this comment, we have also added several additional antigen-specific blood analyses at month 6 post vaccination in the revised manuscript, as covered in responses to other reviewer comments (see revised Figures 4,5 & Supplemental Figure 4).

6. Supplementary Fig 2 – it is strange that no activation markers are observed on T cells in the blood or nasal tissue following vaccination. Can the authors comment on why they think this is occurring? Moreover, can they confirm that in the blood they were gating on antigen experienced (CD45RO+) cells?

We also found this to be surprising and contrary to our hypothesis. We developed this hypothesis based on studies of live attenuated vaccines, which likely have a much larger innate ‘footprint’ compared to the novel mRNA vaccine platforms. One possibility is that mRNA vaccines are more ‘targeted’, and therefore lead to lower magnitude of T cell activation. This question will need to be addressed in future studies.

It is a good point about CD45RO. While this marker was included in our in AIM/ICS panel, it was not included in the kinetics analysis panel due to space constraints.

7. Supplementary Figure 3 – Further work should be done to define the cell type the CD3-CD4-CD8- cells and the CD3+CD4-CD8- pool represent – these cells seem important as they are expanding following vaccination? RNAseq analysis would be useful here as cell numbers are low.

We agree completely with this suggestion, although it is not feasible to do in this study. Because the role of those cells in vaccination is unknown, this would require a dedicated study including the optimization of the RNAseq method for this sample type following cell sorting.

8. Figure 3 – Can a control be performed stimulating T cells recovered from the baseline sample of these donors. Or if limiting, using T cells recovered from unvaccinated individuals would also be useful to show that the observed response is specific. Moreover, can the authors show whether activation markers profiled earlier (ie HLA-DR and CD38) become elevated following stimulation? In addition, it would be useful to show a similar analysis on the matched blood samples.

This is a very good suggestion and certainly something we considered at the start of the study. To maximize cell recovery, we have been doing all of the staining on freshly isolated cells. It was not possible to do both stimulation and enumeration/phenotyping at baseline, and we felt the latter was more appropriate given the novelty of this work. We also considered sampling unvaccinated individuals, but the rapid vaccine rollout has made this extremely challenging. While there are still unvaccinated people, this group may also be unlikely to participate in research studies. We also decided that unstimulated vaccine-recipient samples would be a more stringent control than either of the other possibilities, and that their memory T cells should have gone into a resting state being >2 months post vaccine. Note that the intracellular cytokine/cytotoxicity/CD40L analysis has now been updated (Figures 4 & 5) to include paired stimulated and unstimulated samples.

HLA-DR and CD38 were not included in the stimulation panel as we decided to focus on AIM and cytokines and were limited in terms of colours. Also, because there was no change in kinetics, we decided against further analysis of these markers for this vaccine. Although it is a good point they could increase after stimulation, the AIM markers are better validated for this purpose.

9. Figure 4 – Although total CD8 Trm numbers in the nasal tissue do not correlate with Ab titres perhaps antigen specific CD8 Trm do? It would be interesting to see the % of CD8+TNF+ or CD8+IFN γ + cells post stimulation plotted against Ab titre.

While this is certainly possible to do in future studies, we did not measure antigen-specific CD8+ T cells at the peak antibody time points included in this paper. The association between antibodies (especially mucosal) and T cell responses may need to be the focus of future research.

Minor

- Fig 2 – part F labelled incorrectly in the figure legend

We have corrected this error.

** See Nature Research's author and referees' website at www.nature.com/authors for information about policies, services and author benefits.

Our flexible approach during the COVID-19 pandemic

If you need more time at any stage of the peer-review process, please do let us know. While our

systems will continue to remind you of the original timelines, we aim to be as flexible as possible during the current pandemic.

This email has been sent through the Springer Nature Tracking System NY-610A-NPG&MTS

Confidentiality Statement:

This e-mail is confidential and subject to copyright. Any unauthorised use or disclosure of its contents is prohibited. If you have received this email in error please notify our Manuscript Tracking System Helpdesk team at <http://platformsupport.nature.com>.

Details of the confidentiality and pre-publicity policy may be found here <http://www.nature.com/authors/policies/confidentiality.html>

Privacy Policy | Update Profile

DISCLAIMER: This e-mail is confidential and should not be used by anyone who is not the original intended recipient. If you have received this e-mail in error please inform the sender and delete it from your mailbox or any other storage mechanism. Springer Nature Limited does not accept liability for any statements made which are clearly the sender's own and not expressly made on behalf of Springer Nature Ltd or one of their agents.

Please note that Springer Nature Limited and their agents and affiliates do not accept any responsibility for viruses or malware that may be contained in this e-mail or its attachments and it is your responsibility to scan the e-mail and attachments (if any).

Springer Nature Ltd. Registered office: The Campus, 4 Crinan Street, London, N1 9XW. Registered Number: 00785998 England.

REVIEWERS' COMMENTS

Reviewer #1 (Remarks to the Author):

Thank you for the revisions. This is now much more clear.

In figure 6 - what are the 2 individuals that are indicated with red dots? Can you explain in the figure legend if important.

Figure 6 also has the word 'aint' on the top corner that is probably not in the right place?

As a suggestion - is there a way to show means or median on spaghetti plots, sometimes it is hard to see the trend in these.

Reviewer #2 (Remarks to the Author):

The authors have sufficiently addressed my comments in the revised manuscript and response letter. I have only a text-related comment:

The legend of Figure 6 should explain the meaning of the red symbols (patients with prior COVID-19).

Reviewer #3 (Remarks to the Author):

The authors have addressed most of my concerns. However, I have a few remaining minor concerns that should be addressed.

1. Modify Figure 1c – the Trm gate on CD4 is shown as CD103+CD69+ cells which is misleading and should be modified to include all CD69+ cells. Moreover, now there is no mention in the text (except the figure legend) of the CD161 and CCR6 analysis performed in figure 1. This should be explained in the results section not just the figure legend. Also the label Th17 should be removed from the Figure

2. Figure 2C – remove reference to TRM for blood CD8+CD69+CD103+ cells in the figure legend. Moreover, as the authors did not include CD45RO or other markers of antigen experience it is not appropriate to call these cells in the blood “memory” T cells. Can the authors include a supplementary figure showing their gating strategy these used in their blood analysis – how did they exclude non-conventional T cells etc from their analysis?

Figure 2b-d – can the authors define in the figure legend what the % of cells is relative to (total lymphocytes or total CD8/CD4?). This needs to be mentioned in the text not just the rebuttal letter.

Author's Response

Reviewer #1 (Remarks to the Author):

Thank you for the revisions. This is now much clearer.

In figure 6 - what are the 2 individuals that are indicated with red dots? Can you explain in the figure legend if important?

These are the 2 individuals with prior COVID-19 infection (pre-vaccine), highlighted in response to another reviewer. We have updated the figure legend to ensure this is clear.

Figure 6 also has the word 'aint' on the top corner that is probably not in the right place?

We apologize for this error which has now been fixed, and all figures are now uploaded individually at the end.

As a suggestion - is there a way to show means or median on spaghetti plots, sometimes it is hard to see the trend in these.

For Figure 6 the medians are already shown, however, in many cases the median is at "0" and therefore can't be seen.

Reviewer #2 (Remarks to the Author):

The authors have sufficiently addressed my comments in the revised manuscript and response letter. I have only a text-related comment:

The legend of Figure 6 should explain the meaning of the red symbols (patients with prior COVID-19).

We thank the reviewer for their comment. We agree and have addressed this in response to R1, above.

Reviewer #3 (Remarks to the Author):

The authors have addressed most of my concerns. However, I have a few remaining minor concerns that should be addressed.

1. Modify Figure 1c – the Trm gate on CD4 is shown as CD103+CD69+ cells which is misleading and should be modified to include all CD69+ cells. Moreover, now there is no mention in the text (except the figure legend) of the CD161 and CCR6 analysis performed in figure 1. This should be explained in the results section not just the figure legend. Also the label Th17 should be removed from the Figure

We thank the reviewer for these corrections. We have removed Th17 from the figure legend and have more clearly indicated that all CD69+ CD4+ T cells can be considered

“Trm”.

2. Figure 2C – remove reference to TRM for blood CD8+CD69+CD103+ cells in the figure legend. Moreover, as the authors did not include CD45RO or other makers of antigen experience it is not appropriate to call these cells in the blood “memory” T cells. Can the authors include a supplementary figure showing their gating strategy these used in their blood analysis – how did they exclude non-conventional T cells etc from their analysis?

We have made a new gating strategy for PBMCs in the supplementary data. We have removed the reference to Trm in the figure legend.

Figure 2b-d – can the authors define in the figure legend what the % of cells is relative to (total lymphocytes or total CD8/CD4?). This needs to be mentioned in the text not just the rebuttal letter.

We have updated the denominators here to reflect that the cell populations described are % of CD4+ and CD8+ T cells. This is now indicated in the figure legend.